# Impact of obesity on life expectancy among different European countries: secondary analysis of population-level data over the 1975–2012 period

Nikoletta Vidra,[1] Sergi Trias-Llimós,[1,2] Fanny Janssen[3]

[1]Population Research Centre, Faculty of Spatial Sciences, University of Groningen, Groningen, The Netherlands
[2]Faculty of Epidemiology and Population Health, London School of Hygiene and Tropical Medicine, London, UK
[3]Netherlands Interdisciplinary Demographic Institute, The Hague, The Netherlands

**Correspondence to**
Dr Nikoletta Vidra;
n.vidra@rug.nl

## ABSTRACT

**Objective** This study assesses the impact of obesity on life expectancy for 26 European national populations and the USA over the 1975–2012 period.

**Design** Secondary analysis of population-level obesity and mortality data.

**Setting** European countries, namely Austria, Belarus, Belgium, the Czech Republic, Denmark, Estonia, Finland, France, Hungary, Iceland, Ireland, Italy, Latvia, Lithuania, Luxembourg, the Netherlands, Norway, Poland, Portugal, the Russian Federation, Slovakia, Spain, Sweden, Switzerland, Ukraine and the UK; and the USA.

**Participants** National populations aged 18–100 years, by sex.

**Measurements** Using data by age and sex, we calculated obesity-attributable mortality by multiplying all-cause mortality (Human Mortality Database) with obesity-attributable mortality fractions (OAMFs). OAMFs were obtained by applying the weighted sum method to obesity prevalence data (non-communicable diseases (NCD) Risk Factor Collaboration) and European relative risks (Dynamic Modeling for Health Impact Assessment (DYNAMO-HIA)). We estimated potential gains in life expectancy (PGLE) at birth by eliminating obesity-attributable mortality from all-cause mortality using associated single-decrement life tables.

**Results** In the 26 European countries in 2012, PGLE due to obesity ranged from 0.86 to 1.67 years among men, and from 0.66 to 1.54 years among women. In all countries, PGLE increased over time, with an average annual increase of 2.68% among men and 1.33% among women. Among women in Denmark, Switzerland, and Central and Eastern European countries, the increase in PGLE levelled off after 1995. Without obesity, the average increase in life expectancy between 1975 and 2012 would have been 0.78 years higher among men and 0.30 years higher among women.

**Conclusions** Obesity was proven to have an impact on both life expectancy levels and trends in Europe. The differences found in this impact between countries and the sexes can be linked to contextual factors, as well as to differences in people's ability and capacity to adopt healthier lifestyles.

## INTRODUCTION

Obesity is a global epidemic,[1] with Europe currently ranking second worldwide after

### Strengths and limitations of this study

► This is the first study to assess the impact of obesity on life expectancy at birth over time, and we do so here for 26 European countries and the USA.
► We used recent long-term comparable data on obesity prevalence based on population-based measurement studies, and European relative risks of dying from obesity by age and sex from a recent meta-analysis.
► Because of remaining data limitations regarding prevalence and relative risks, we had to use a fairly simple—although commonly applied—methodology to estimate obesity-attributable mortality.

the USA.[2] Over the last 20 years obesity prevalence has increased threefold in Europe,[3] although not uniformly across countries.[4] Estimates for 2014 indicate that obesity varied threefold across European countries, ranging from a low of 9% in Romania to a high of 26% in Malta.[5] Obesity constitutes a serious health burden at the individual and population levels because it is associated with an increased risk of morbidity[6] and mortality.[7] However, the potential impact of the increase in obesity on life expectancy trends remains largely unknown.[8]

The few existing studies that assessed the impact of obesity on life expectancy at the population level provided estimates at one specific point in time only.[9 10] Olshansky *et al*[9] found that if obesity was eliminated, life expectancy at birth ($e_0$) in the USA in 2000 would be 0.21–1.08 years higher, depending on gender and ethnicity. Preston *et al*[10] estimated for 16 low-mortality countries in 2006 that the reduction in life expectancy at age 50 ($e50$) due to obesity was greatest in the USA, at more than 1.5 years, and ranged from 0.50 to 1.19 years for women and from 0.72 to 1.37 years for men in European countries.

Gaining insight into the impact of obesity on trends in life expectancy is especially relevant[11] given the marked differences in life expectancy trends across Europe.[12] In Western European countries, $e_0$ has been increasing steadily, and has risen 6–8 years since 1970. But in Central and Eastern Europe (CEE), $e_0$ stagnated or even declined between the 1970s and the 1980s, and did not start increasing again until the 1990s. There are also marked differences in $e_0$ trends between individual European countries.[12]

In light of these important differences between European countries in both obesity prevalence and life expectancy over time, our aim is to assess the impact of obesity on long-term trends in life expectancy across a wide range of European countries.

Our sole focus is on the impact of obesity, given the significant health burden caused by obesity, the large body of literature on its impact and the well-documented association of obesity with mortality.

## DATA AND METHODS
### Setting
We studied the impact of obesity on life expectancy by sex over the 1975–2012 period in 26 European countries: Austria, Belarus, Belgium, the Czech Republic, Denmark, Estonia, Finland, France, Hungary, Iceland, Ireland, Italy, Latvia, Lithuania, Luxembourg, the Netherlands, Norway, Poland, Portugal, the Russian Federation, Slovakia, Spain, Sweden, Switzerland, Ukraine and the UK; and the USA as a comparison country.[10]

### Data
Long-term comparable obesity prevalence data (body mass index (BMI) ≥30 kg/m$^2$) by country, sex, age (18–19, 20–24, …, 85+) and year (1975–2012), based on 1698 population-based measurement studies, were obtained from the non-communicable diseases (NCD) Risk Factor Collaboration study.[13] These validated data comprise the available measured height and weight data of 19.2 million participants from representative data sources, supplemented with estimates based on information from other years and related countries from a Bayesian hierarchical model.[13] The same model was applied to all countries and used as an input in the measured weight and height data, including covariates that help predict BMI.[13]

The age-specific (<50, 50–59, 60–69 and ≥70 years) and sex-specific relative risks (RRs) of dying from obesity (see online supplementary table S1) came from a review of studies mainly conducted in Western Europe and the USA, with the normal-weight category used as the reference group (18.5≥BMI≤25 kg/m$^2$).[14] These age-specific and sex-specific RRs were largely in line with the overall European RR of 1.64 recently estimated by the Global BMI Mortality Collaboration.[7] The differences across age groups found in that study were similar with those reported in our findings (ie, higher RRs at younger than at older ages), although they were less distinct.[7] In

addition, the use of RRs with the normal-weight category as the reference category is in line with previous studies that estimated obesity-attributable mortality,[15–19] while the estimation of obesity-attributable mortality with such an RR can be considered the theoretically maximally possible attributable mortality.[20]

All-cause mortality numbers and exposure population data by single year of age, sex, year and country were obtained from the Human Mortality Database.[21] These data are of high quality and are widely used within the demographic community and beyond.[22]

### Patient and public involvement
No patients were involved in this study.

## METHODS
We performed our analyses separately by country and sex, based on data by single year of age (18–100). The obesity prevalence data were turned into single-age prevalence (18–100) by applying Loess smoothing.[23] The RRs were turned into single-year RRs (18–100) using linear regression.

To estimate the obesity-attributable mortality fraction (OAMF)—that is, the share of all-cause mortality due to obesity—we used the Rockhill formula to estimate OAMFs by age ($a$) and sex ($s$).[24]

$$OAMF_{a,s} = \frac{P_{a,s} \cdot (RR_{a,s}-1)}{1+(P_{a,s} \cdot (RR_{a,s}-1))} \text{ (Equation 1)}$$

where $P$ is the obesity prevalence. We then weighted the $OAMF_{a,s}$ with the corresponding number of deaths.

For the estimation of the impact of obesity on life expectancy, we needed age-specific and sex-specific (non-)obesity-attributable mortality rates. These were obtained by multiplying $OAMF_{a,s}$ and [1– $OAMF_{a,s}$], respectively, with age-specific and sex-specific all-cause mortality rates.

To ensure comparability across countries, over time, and between men and women, we applied direct age and sex standardisation[25] to obesity prevalence, OAMFs and obesity-attributable mortality rates, using the European population of 2011[26] as the standard.

To assess the impact of adult obesity on $e_0$, we calculated for each country the potential gain in life expectancy (PGLE) if obesity-attributable mortality was eliminated, by calendar year and sex, and is in line with the approach by Preston *et al*.[10] First, we calculated $e_0$ by applying standard life table techniques to age-specific all-cause mortality rates (0–100)[25]). Second, we applied associated single-decrement life tables (ASDLT[25]) to age-specific and sex-specific non-obesity-attributable mortality rates (0–100) to obtain $e_0$ if obesity-attributable mortality was eliminated. The PGLE represents the difference between the $e_0$ based on the ASDLT and the original $e_0$.

To summarise the changes in PGLE across countries, we estimated the average annual changes in PGLE (in %):

$$Average\ annual\ changes\ in\ PGLE\ (\%) = \frac{\sum_{t=1976}^{2012} \left( \frac{PGLE_t - PGLE_{t-1}}{PGLE_{t-1}} \right)}{2012-1975} \ 100$$

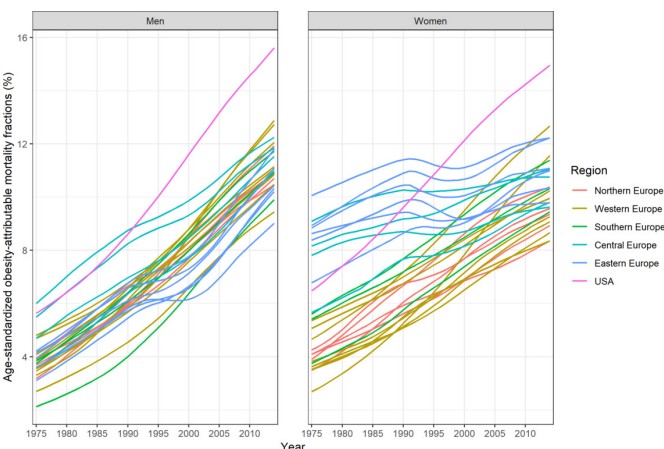

**Figure 1** Age-standardised obesity-attributable mortality fractions in 26 European countries* (by five regions) and the USA, 1975–2014, 18–100 years. *Countries within the same region are presented with the same colour. Central Europe: the Czech Republic, Hungary, Poland and Slovakia; Eastern Europe: Belarus, Estonia, Ukraine, Latvia, Lithuania and the Russian Federation; Northern Europe: Denmark, Finland, Iceland, Norway and Sweden; Southern Europe: Italy, Portugal and Spain; Western Europe: Austria, Belgium, France, Ireland, Luxembourg, the Netherlands, Switzerland and the UK.

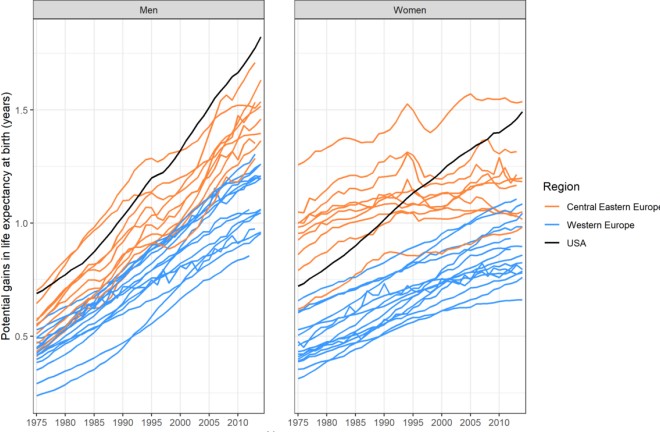

**Figure 2** Potential gains in life expectancy at birth if obesity-attributable mortality was eliminated, in 26 European countries* (differentiating Western and Central Eastern Europe) and the USA, 1975–2012. *Countries within the same region are presented with the same colour. Central Eastern Europe: Belarus, the Czech Republic, Estonia, Hungary, Latvia, Lithuania, Poland, the Russian Federation, Slovakia and Ukraine; Western Europe: Austria, Belgium, Denmark, Finland, France, Iceland, Ireland, Italy, Luxembourg, the Netherlands, Norway, Portugal, Spain, Sweden, Switzerland and the UK.

To assess the impact of obesity on time trends in $e_0$ between 1975 and 2012, we subtracted the observed change in $e_0$ from the change in $e_0$ without obesity. The change in $e_0$ without obesity was obtained by using the $e_0$ values from the ASDLT applied to non-obesity-attributable mortality for 1975 and 2012.

## RESULTS

For the 26 European countries, the age-standardised OAMF was, on average, 11% among men and 10% among women in 2012. For the USA, these estimates were substantially higher, that is, 15% and 14%, respectively. The average OAMF levels were higher in Northern, Western and Southern Europe combined (hereafter, Western Europe) than in CEE among men, while the opposite was the case among women.

OAMFs were increasing over time for all countries and both sexes, although not to the same extent (see figure 1, online supplementary figure S1). In Western Europe, OAMFs generally increased over the 1975–2012 period, and at a faster pace among men. In CEE, by contrast, OAMFs clearly stagnated, and even declined between 1990 and 2000. The overall increase in OAMFs was greatest in the USA, Ireland, Norway (men) and the UK (women).

In the 26 European countries in 2012, estimates of PGLE at birth if obesity was eliminated ranged from 0.86 to 1.67 years among men (1.22 on average) and from 0.66 to 1.54 years among women (0.98 on average) figure 2, online supplementary figure S2 and table S2). Among men in the USA, the PGLE estimate was, at 1.73 years, slightly higher than the highest estimate in Europe;

and among women in the USA, the PGLE estimate was, at 1.44 years, the second highest after the estimate for Russia. The average PGLE estimate was 1.08 among men and 0.86 among women in Western Europe, and was 1.44 among men and 1.16 among women in CEE (see online supplementary table S2).

Overall, from 1975 to 2012, PGLE due to obesity increased in all of the countries (figures 2–4, online supplementary figure S2). The increase was greater among men (average annual increase of 2.68%) than among women (average annual increase of 1.33%), was largest among men in Portugal and Belarus and among women in Portugal, and was substantial among men and women in Norway (figures 3 and 4). While there was a general increase in PGLE due to obesity, this trend stagnated among women in CEE from around 1990 onwards, and levelled off after 1995 among women in Denmark and Switzerland.

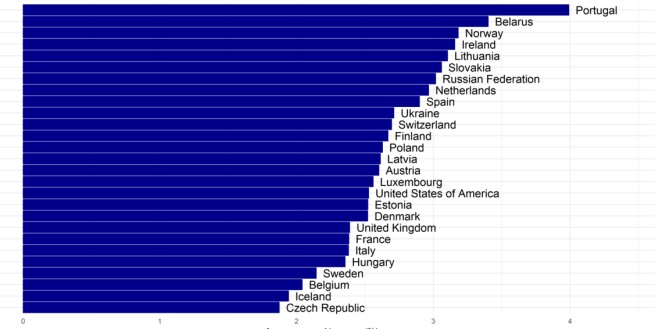

**Figure 3** Average annual increase (%) in potential gains in life expectancy due to obesity in 26 European countries and the USA between 1975 and 2012, in men.

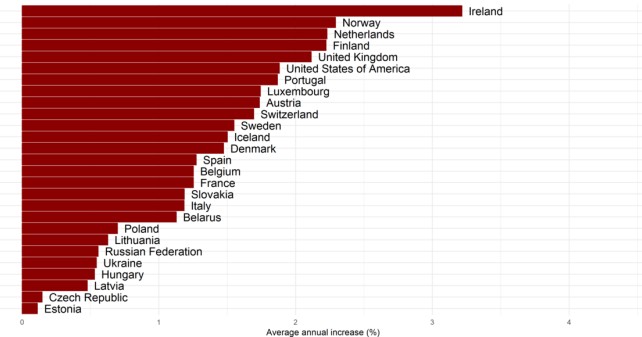

**Figure 4** Average annual increase (%) in potential gains in life expectancy due to obesity in 26 European countries and the USA between 1975 and 2012, in women.

Table 1 shows the impact of obesity on time trends in life expectancy at birth ($e_0$). Overall, the average increase in $e_0$ between 1975 and 2012 was 7.26 years for men and 6.28 years for women in the 26 European countries. Without obesity, the average increase in $e_0$ would have been 8.04 years for men and 6.58 years for women, or 0.78 and 0.30 years higher, respectively. Among men, obesity had the greatest impact on $e_0$ trends in Lithuania and the USA (more than 1 year), and the smallest impact in Iceland and Sweden (0.5 years). Among women, obesity had the greatest impact on $e_0$ trends in the USA and Ireland (0.7 years) and the smallest impact in Estonia and the Czech Republic (less than 0.1 year).

## DISCUSSION
### Summary of results
In the 26 European countries studied, the share of mortality due to obesity in 2012 was, on average, 11% among men and 10% among women. PGLE due to obesity in 2012 ranged from 0.86 to 1.73 years among men, and from 0.66 to 1.54 years among women. Overall, PGLE increased between 1975 and 2012, although more quickly among men (average annual increase: 2.68%) than among women (1.33%). Among women in Denmark, Switzerland and the CEE countries, the increase in PGLE levelled off after 1995. Without obesity, the average increase in $e_0$ between 1975 and 2012 would have been 0.78 years higher among men and 0.30 years higher among women.

### Evaluation of data and methods
Using the recent advances in obesity data, it is now possible to study the impact of obesity on life expectancy for a large number of countries and a long period of time. Two methodological issues warrant our attention, however.

First, in calculating the share of mortality due to obesity (OAMF), which also forms the basis for our PGLE calculations, we were hindered by limitations in the available prevalence and RRs data, which also affected the method used. As has previously been documented, OAMF estimates are sensitive to the data and the methods used.[27]

In selecting obesity prevalence data, we used the longest validated time series based on population-based measurement studies that are suitable for studying the impact of obesity on long-term life expectancy trends across Europe.[13] For those countries with less available obesity data—especially the CEE countries—a portion of the data we used were merely the result of modelling. Thus, the resulting estimates should be treated with some caution. By contrast, for the non-CEE countries, most of the data we used pertain to measured data.[13] Online supplementary table S3 gives the confidence intervals around the age-standardised prevalence estimates for each country by sex in order to provide more information on the relative reliability of the data for the different countries in our analysis.

Because age-specific and sex-specific RRs of mortality associated with obesity are not readily available by country and year, we have decided to apply to all of the countries studied age-specific and sex-specific RRs from Western European and US populations that are largely suitable for our setting, as had previously been done.[10] Although RRs could differ slightly across contexts, studies that compared RRs across continents found only small differences in RRs between Europe and North America.[7] Consequently, we do not expect to observe large differences between individual countries. In addition, as time-variant European RRs were not available, we had to apply time-constant RRs, even though it is possible that changes in the association of obesity with mortality—which could, for example, occur because of improvements in the treatment of chronic diseases—have affected the impact of obesity on life expectancy. Previous studies that assessed changes over time in the association of obesity with mortality did so only for the USA, and unfortunately provided mixed evidence, with some of these studies reporting a decline[18 28 29] and others finding an increase.[30] Therefore, before implementing time-variant European RRs, more information on their direction is required. Similarly, comparable country-specific RRs are urgently needed. In addition, the choice of these RRs along with their reference group might exert an effect on our estimates. Based on the available data, only a fairly simple method—although commonly applied namely the weighted sum method could be applied[27] to estimate the OAMFs. The application of a more advanced methodology[27] could have affected the OAMFs and thus the PGLE levels, but less the trends.[31] The lack of information on the uncertainty of the RRs we used limited us in estimating CIs for the OAMFs and PGLEs.

Second, besides being the result of the OAMFs, PGLE estimates can also be affected by all-cause mortality levels and trends as age-specific and sex-specific all-cause mortality rates are used to estimate PGLE. Since all-cause mortality fluctuated greatly in CEE in the analysed period,[12] short-term variations in PGLE in CEE countries should be treated with more caution.

**Table 1** Impact of obesity on trends in life expectancy at birth ($e_0$) in 26 European countries and USA in 1975–2012, by sex

| Country | Change in $e_0$ with obesity, 1975–2012 (years) | | Change in $e_0$ without obesity, 1975–2012 (years) | | Effect of obesity on $e_0$ change, 1975–2012 (years) | |
|---|---|---|---|---|---|---|
| | Men | Women | Men | Women | Men | Women |
| Austria | 10.62 | 8.61 | 11.25 | 8.95 | 0.63 | 0.34 |
| Belarus | −0.55 | 1.43 | 0.46 | 1.83 | 1.00 | 0.40 |
| Belgium | 8.85 | 7.63 | 9.46 | 7.99 | 0.61 | 0.36 |
| Czech Republic | 7.97 | 6.98 | 8.66 | 7.03 | 0.69 | 0.05 |
| Denmark | 6.78 | 5.03 | 7.40 | 5.36 | 0.63 | 0.33 |
| Estonia | 6.43 | 6.42 | 7.26 | 6.46 | 0.82 | 0.04 |
| France | 9.49 | 7.99 | 10.17 | 8.30 | 0.68 | 0.31 |
| Finland | 10.07 | 7.26 | 10.82 | 7.75 | 0.74 | 0.50 |
| Hungary | 5.29 | 6.18 | 6.17 | 6.36 | 0.87 | 0.18 |
| Iceland | 9.02 | 5.19 | 9.51 | 5.51 | 0.48 | 0.32 |
| Ireland | 9.40 | 8.40 | 10.22 | 9.10 | 0.83 | 0.69 |
| Italy | 10.19 | 8.56 | 10.81 | 8.89 | 0.62 | 0.33 |
| Latvia | 4.91 | 4.53 | 5.82 | 4.70 | 0.90 | 0.18 |
| Lithuania | 2.01 | 3.80 | 3.14 | 4.06 | 1.13 | 0.26 |
| Luxembourg | 11.78 | 9.27 | 12.50 | 9.65 | 0.72 | 0.37 |
| The Netherlands | 7.68 | 5.10 | 8.26 | 5.6 | 0.56 | 0.49 |
| Norway | 7.70 | 5.33 | 8.42 | 5.86 | 0.74 | 0.51 |
| Poland | 5.90 | 6.74 | 6.81 | 7.00 | 0.91 | 0.27 |
| Portugal | 12.14 | 10.87 | 12.91 | 11.26 | 0.77 | 0.40 |
| Russian Federation | 2.05 | 2.62 | 3.06 | 2.89 | 1.02 | 0.28 |
| Slovakia | 5.65 | 5.82 | 6.52 | 6.16 | 0.88 | 0.34 |
| Spain | 8.82 | 8.75 | 9.62 | 9.14 | 0.79 | 0.39 |
| Sweden | 7.69 | 5.59 | 8.18 | 5.93 | 0.49 | 0.33 |
| Switzerland | 8.98 | 6.63 | 9.55 | 6.93 | 0.58 | 0.30 |
| Ukraine | 0.48 | 1.73 | 1.26 | 1.94 | 0.78 | 0.21 |
| UK | 9.46 | 6.96 | 10.20 | 7.55 | 0.74 | 0.59 |
| USA | 7.86 | 4.89 | 8.90 | 5.61 | 1.04 | 0.71 |
| Average Central and Eastern European countries | 4.01 | 4.63 | 4.92 | 4.84 | 0.90 | 0.22 |
| Average Western countries | 9.10 | 7.09 | 9.76 | 7.50 | 0.66 | 0.41 |

### Explanation of results

In 2012, the PGLE due to obesity were, on average, 1.22 years for men and 0.98 years for women in the 26 European countries, and 1.73 years for men and 1.43 years for women in the USA. A comparison of our 2006 e50 estimates with those of Preston et al[10] for the same countries uncovered only small differences, except among men in the USA (our estimate was 0.56 years lower) and women in the UK (our estimate was 0.29 years lower) (see online supplementary table S4). Given that approximately the same methodology was used to estimate the OAMFs, the observed differences are most likely due to the use of different obesity prevalence and RRs data. Preston et al[10] used prevalence data from national representative surveys and RRs from the prospective studies collaboration. Given that the observed differences do not have the same direction for the different countries, we believe that these differences are mainly attributable to the prevalence data used. To further evaluate our observed PGLE levels, we compared them with own PGLE estimates for smoking and alcohol.[32] Our PGLE estimates for smoking were 2.38 years for men and 1.00 year for women in Western Europe, and 3.82 years for men and 0.67 years for women in CEE. Our PGLE estimates for alcohol were 0.90 years for men and 0.44 years for women in Western Europe, and 2.15 years for men and 1.00 year for women in CEE,[32] while our average PGLE for obesity was 1.08 among men and 0.86 among women in Western Europe, and 1.44 among men and 1.16 among women in CEE. Thus, obesity's impact on life expectancy lies between that of smoking and alcohol, and can be considered significant.

In our study, we found that PGLE due to obesity was increasing, but that this trend differed across countries and between the sexes. This overall trend can be explained by the general increase in obesity prevalence in European countries (see online supplementary figure S3)[13] and the resulting growth in the burden of obesity,[3] which is also reflected in the OAMFs (figure 1, online supplementary figure S1) in these countries.

At the same time, parts of the observed variation in the increase in PGLE estimates across the USA, Western Europe and CEE and between the sexes reflect differences in the onset, the development and the impact of the obesity epidemic in these countries and in men and women. Across the countries studied, the absolute increase in PGLE was largest among women and the second largest among men in the USA. This pattern is in line with evidence showing that between 1980 and 2008, obesity increased much more in the USA than in Europe.[1 33] This rapid progression of the obesity epidemic in the USA and its large impact on life expectancy has been attributed to an increasingly obesogenic environment caused by factors such as changes in food preparation and processes that promote the consumption of calorically dense foods, and a pronounced decrease in physical activity levels.[34] The obesity epidemic has progressed more slowly in Western Europe than in the USA.[1 13] However, obesity levels in countries like the UK

and Ireland are rapidly approaching those in the USA,[35] as our PGLE estimates also show.

In the CEE countries, the PGLE trends track the evolution of the obesity epidemic in that region (see online supplementary figure S3). Obesity levels have been higher in CEE than in Western Europe since 1980,[36 37] which suggests that the epidemic started earlier in CEE. As a result of this earlier onset, the impact of obesity (as expressed in terms of OAMF and PGLE) in the 1970s and 1980s was at times even greater in CEE than in the USA, especially among women. While there are many potential explanations for this early onset of the obesity epidemic in CEE, the available data indicate that the main factors were the relatively high total energy supply and energy intake in CEE in those years.[38]

The overall progress of the obesity epidemic was lower in CEE than in Western Europe, and the increase was not constant.[1] Indeed, in CEE, increases in obesity prevalence,[1 39] OAMFs and PGLE stagnated in the 1980–2008 period, and were more pronounced in the 1990s.[1 39] However, as these countries started the study period with higher obesity prevalence levels, these trends resulted in the CEE having higher average PGLE levels than the West. The pattern of stagnation observed in the CEE could be explained by the decrease in energy supplies at the beginning of the 1990s in CEE,[37] resulting from the dramatic economic and political changes in those countries,[3 36 39] and which in turn affected the socioeconomic status (SES) of the population. Among CEE women, the increase in obesity starting in the 1990s was smaller than it was in the previous period and was smaller than it was among CEE men. The lower risk of obesity observed among women than among men with low SES in low-income countries[40] may explain this difference.

In Western Europe, a stagnation in PGLE levels was observed among women in Denmark and Switzerland after 1995. This finding seems to be in line with studies reporting a levelling off of mean BMI since the 1990s,[41] and in specific subpopulations such as adults with high SES in regions within Switzerland, Italy, France and Finland.[37] Although dietary and physical activity information is spreading equally across socioeconomic groups, those with higher SES have a greater ability and capacity to adopt a healthier dietary and physical activity pattern.[42] In addition, it appears that higher SES women in particular are more health-conscious, have healthier food habits and are more prone to follow nutritional recommendations[43] as they are under greater social pressure to be thin.[44] Similarly, countries with higher income levels and lower levels of inequality,[45] like Switzerland and Denmark, tend to have lower obesity levels, especially among women.

When we considered the impact of obesity on life expectancy in the 26 European countries, we found that without obesity, the increase in $e_0$ between 1975 and 2012 would have been, on average, 0.78 years higher among men and 0.30 years higher among women. These figures account for approximately 10% of the average change in

$e_0$ between 1975 and 2012 among men and 5% among women. It is therefore clear that the impact of obesity on changes in $e_0$ should not be ignored. Moreover, the impact of obesity on life expectancy levels and on life expectancy trends is likely to increase, as previous studies have also suggested.[8] There are several indicators pointing in that direction, including evidence that obesity's impact is already substantially greater in the USA (13% among men and 15% among women) than elsewhere; obesity prevalence is increasing rapidly in most European countries (see online supplementary figure S3); obesity is increasing in severity; and the duration of obesity is rising in younger generations.[8]

## CONCLUSION AND IMPLICATIONS

Obesity was proven to have an impact on both life expectancy levels and trends in Europe. The observed differences in the increase in the impact of obesity across countries and between the sexes reflect differences in the onset and the progression of the obesity epidemic, and can be linked to contextual factors (economic conditions, obesogenic environment, energy supplies), as well as to differences in people's ability and capacity to adopt healthier lifestyles.

It is likely that in the future obesity will have a larger impact on mortality and life expectancy in Europe, as obesity prevalence and obesity-attributable mortality continue to increase in the majority of countries. These trends will have important health, economic and social implications. Specifically, the increasing prevalence of obesity among European populations, and especially at younger ages, will lead to an increased prevalence of obesity-related disorders, as well as to increases in the mortality burden associated with obesity and in obesity's effects on life expectancy and quality of life. Thus, obesity will constitute an additional burden for societies, economies and public health. It is therefore crucial that effective public health initiatives are undertaken to tackle the obesity epidemic and its effects on public health. Such initiatives should address the multifactorial and complex obesity aetiology, the clear differences between countries and the sexes, as well as the factors underlying these differences, such as contextual factors and differences in individuals' ability and capacity to adopt healthier lifestyles.

**Acknowledgements** The authors are grateful to Professor Majid Ezatti and Dr James Bentham (Faculty of Medicine, School of Public Health at Imperial College London), Dr Gretchen A Stevens (WHO), and the NCD Risk Factor Collaboration (NCD-RisC) (www.ncdrisc.org) for sharing the obesity data.

**Contributors** NV contributed to the conception and design of the work; to the acquisition, analysis and interpretation of data for the work; drafted and revised the work critically for important intellectual content; approved the final version of the work to be published; and agrees to be accountable for all aspects of the work in ensuring that questions related to the accuracy or integrity of any part of the work are appropriately investigated and resolved. ST-L contributed to the analysis and interpretation of data for the work; revised critically the final draft for important intellectual content; approved the final version of the work to be published. FJ contributed to the conception and design of the work; contributed to the interpretation of data for the work; revised the work critically for important intellectual content; approved the final version of the work to be published.

**Funding** This work was funded by the Netherlands Organisation for Scientific Research (NWO) in connection with the research programme 'Smoking, alcohol, and obesity, ingredients for improved and robust mortality projections', grant no 452-13-001 (see www.futuremortality.com).

**Competing interests** None declared.

**Patient consent for publication** Not required.

**Provenance and peer review** Not commissioned; externally peer reviewed.

**Data sharing statement** Obesity prevalence data can be obtained from NCD Risk Factor Collaboration (NCD-RisC) (www.ncdrisc.org). All-cause mortality and exposure population data can be obtained from the Human Mortality Database (http://www.mortality.org).

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
