## [Reviewer comments · BMJ Open]

ARTICLE DETAILS

TITLE (PROVISIONAL)	Impact of obesity on life expectancy among different European countries: secondary analysis of population-level data over the 1975-2012 period
AUTHORS	Vidra, Nikoletta; Trias-Llimós, Sergi; Janssen, Fanny

VERSION 1 – REVIEW

REVIEWER	Lena Björck Department of Molecular and Clinical Medicine Sahlgrenska Academy at the University of Gothenburg, Sweden
REVIEW RETURNED	11-Dec-2018

GENERAL COMMENTS	Thank you for an interesting paper. My major concern is that the results probably underestimate the effect of weight. Why was this cut-off chosen (BMI ≥ 30) and not BMI ≥ 25? Earlier research has shown that the risk increase at BMI 22.5 for some diseases as weight is a continuous variable. Would it be possible to estimate the effect with a lower cut-off (≥ 25)? Or, if not, to do a simulation? Since the non-obese population probably includes individuals with obesity related disorders the risk is probably higher compared to individuals with lower BMI and the impact on life expectancy underestimated. These issues are of clinical relevance and should be discussed. To discuss the future implications of increasing obesity would strengthen the paper.
---

REVIEWER	Klodian Dhana Rush University Medical Center, United States
REVIEW RETURNED	14-Dec-2018

GENERAL COMMENTS	In this manuscript, the authors aimed to assess the impact of obesity in life expectancy across 26 European countries. The authors estimated the potential gain in life expectancy if obesity-attributable mortality were eliminated using the prevalence data from NCD risk factor collaboration and relative risk from a systematic review conducted in Western Europe and the US. The analysis is novel, relevant, and the manuscript is well-written. I have a few suggestions: 1. Epidemiological studies have shown distinct differences regarding the contribution of overweight and obesity in mortality among young adults, middle-aged, and elderly. For example, data shows that the
--

	impact of obesity on mortality diminish with increasing age. How the authors addressed those differences in their analysis? 2. The effect of obesity on mortality is mediated through chronic diseases, such as cardiovascular disease. In the recent decade, there is an improvement in the treatment of cardiovascular disease, and more individuals are living with cardiovascular disease and contributing to increased life expectancy. Therefore, the contribution of obesity to mortality could be smaller due to better treatment of chronic diseases. I think authors should discuss in the discussion to explain the smaller estimates compared to the previous study (Preston et al.). 3. What is the contribution of the differences in socioeconomic status across 26 European countries, especially comparing East and West Europe to the results?
--	---

REVIEWER	Paola Zaninotto UCL
REVIEW RETURNED	11-Feb-2019

GENERAL COMMENTS	I have read the paper and although I thought that I could provide my expertise in revising it, I have realised that the methods used are not familiar to me, and although I tried to read about them I felt that the statistical analysis was not sufficiently described for me to assess the appropriateness. I feel that a demographer might be better placed in reviewing this article. I am really sorry for not being able to help as much as I would have liked on this occasion.
--

REVIEWER	Haomiao Jia Columbia University, USA
REVIEW RETURNED	01-Mar-2019

GENERAL COMMENTS	This study assessed the population level burden of disease associated with obesity by estimating potential gains in life expectancy by eliminating obesity-attributable mortality from all-cause mortality for 27 countries from 1975 to 2012. This study used data from difference sources for the calculations: (1) prevalence of obesity (BMI\geq30kg/m²), by age, sex, county, and year; (2) relative risk (RR) of dying from obesity, by age categories and sex; and (3) all-cause mortality, by age, sex, and year. The method for the calculation of potential gains in life expectancy is a valid method. The validity and reliability of estimates depend on the data used in this study. More detailed description of data sources is needed. 1. All-cause mortality estimation: All-cause mortality data by age and sex were used for the estimation of life expectancy (life table method). The authors said that this is "single year of age, sex, and year". The authors did not mention whether this is by counties or combined data for all counties. If this is for each counties, what is reliability of the data? Particularly for some small counties. If this is combined data for all counties, further discussion of this weakness is needed. Since life expectancy varied greatly across European counties (approximately 10 years from the lowest to the highest), I wonder if you can use combined data.
--

	2. Obesity prevalence estimates: What data were used for the estimation? The estimates were model based. Different models for different counties, or a single model for all counties? What are predictors for these estimates? 3. RR of dying from obesity: Is obese persons relative to not obese persons or obese persons relative normal weight persons? Many studies estimated relative risk (and sometimes, hazard ratio) of dying of obese persons relative to normal weight persons. If RR is obese persons relative to normal weight person, the estimates are not accurate. Please confirm that RR is relative to not obese persons. 4. Another weakness is applying same RRs to all 27 countries from 1975 to 2012. Many factors (including race/ethnicity and geographic regions) were associated with the impact of obesity on morality. It might be better applying different RRs for different counties (at least for select counties where such data are available). Other 1. Reliability of estimates: This study did not provide standard error or confidence limits of estimates. This information might be important and should be reported (see my comment on reliability life expectancy estimation for small counties above). 2. Figures are different to see.
--	--

VERSION 1 – AUTHOR RESPONSE

Reviewer: 1

Reviewer Name: Lena Björck

Institution and Country: Department of Molecular and Clinical Medicine

Sahlgrenska Academy

at the University of Gothenburg, Sweden

Reviewer's Comments to Author

Comment 1: Thank you for an interesting paper. My major concern is that the results probably underestimate the effect of weight. Why was this cut-off chosen (BMI ≥ 30) and not BMI ≥ 25 ? Earlier research has shown that the risk increase at BMI 22.5 for some diseases as weight is a continuous variable. Would it be possible to estimate the effect with a lower cut-off (≥ 25)? Or, if not, to do a simulation? Since the non-obese population probably includes individuals with obesity related disorders the risk is probably higher compared to individuals with lower BMI and the impact on life expectancy underestimated. These issues are of clinical relevance and should be discussed.

Reply

Yes, it would be possible to estimate the overall effect of weight on life expectancy by choosing a different cut-off and implementing different RRs in the calculations. In our study, however, we focused specifically on the impact of obesity (BMI ≥ 30 kg/m²) on life expectancy. There are two reasons why we did not assess the overall effect of weight; and, consequently, did not include overweight (BMI 25-30 kg/m²) in our analysis. First, we were particularly interested in assessing the impact of obesity, as the prevalence of obesity is alarmingly high, and obesity represents a significant health burden. Given that obesity is one of the biggest public health challenges facing countries today, this issue has attracted extensive attention in the literature. Second, the existing findings on the relationship between overweight and the risk of mortality are not yet conclusive, with some studies reporting that overweight people have a lower relative risk of mortality than the normal weight reference group (Flegal et al., 2013; Flegal et al., 2018) and other studies reporting that the relative risk of mortality is higher among overweight than normal weight people (Aune et al., 2016; Global BMI Mortality Collaboration et al., 2016). In order to clarify our sole interest in obesity, we added the following in the introduction, page 5, lines 90-92: “Our sole focus is on the impact of obesity, given the significant health burden caused by obesity, the large body of literature on its impact, and the well-documented association of obesity with mortality”.

Comment 2: To discuss the future implications of increasing obesity would strengthen the paper.

Reply

We extended our discussion of the likely future implications of a continued increase in obesity in our discussion of our results on page 18, lines 368- 374: “Moreover, the impact of obesity on life expectancy levels and on life expectancy trends is likely to increase, as previous studies have also suggested (Alley et al., 2011). There are several indicators pointing in that direction, including evidence that obesity’s impact is already substantially greater in the USA (13% among men and 15% among women) than elsewhere; obesity prevalence is increasing rapidly in most European countries (see Supplementary Material Figure S3); obesity is increasing in severity; and the duration of obesity is rising in younger generations (Alley et al., 2011)”.

Moreover, on page 18-19, lines 383-391, in our conclusions section, we stressed this point again: “It is likely that in the future obesity will have a larger impact on mortality and life expectancy in Europe, as obesity prevalence and obesity-attributable mortality continue to increase in the majority of countries. These trends will have important health, economic, and social implications. Specifically, the increasing prevalence of obesity among European populations, and especially at younger ages, will lead to an increased prevalence of obesity-related disorders, as well as to increases in the mortality burden associated with obesity and in obesity’s effects on life expectancy and quality of life. Thus, obesity will constitute an additional burden for societies, economies, and public health”.

Reviewer: 2

Reviewer Name: Klodian Dhana

Institution and Country: Rush University Medical Center, United States

Reviewer's Comments to Author

In this manuscript, the authors aimed to assess the impact of obesity in life expectancy across 26 European countries. The authors estimated the potential gain in life expectancy if obesity-attributable mortality were eliminated using the prevalence data from NCD risk factor collaboration and relative risk from a systematic review conducted in Western Europe and the US.

The analysis is novel, relevant, and the manuscript is well-written. I have a few suggestions:

Comment 1: Epidemiological studies have shown distinct differences regarding the contribution of overweight and obesity in mortality among young adults, middle-aged, and elderly. For example, data shows that the impact of obesity on mortality diminish with increasing age. How the authors addressed those differences in their analysis?

Reply

Previous research has indeed shown that the association of BMI with mortality differs across ages; and, specifically, that the relative risk of obesity associated with mortality is higher at younger ages. To account for this pattern, we deliberately selected age- (and sex-) specific relative risks of dying from obesity from a recent meta-analysis (see Table S1, Supplementary Material of the manuscript). After applying linear regression to turn these wide age group RRs into single-year RRs (18- 100), we effectively used RRs declining from 1.53 (age 18) to 1.43 (age 100) for women, and from 1.57 and 1.48 for men, respectively (see the Appendix of this document, Figure 1). Although more recent studies reported greater differences in RRs between age groups (Global BMI Mortality Collaboration et al., 2016), these studies did not, unfortunately, provide sex-specific RRs, which were important for our study as well.

We clarified this point in the text in page 6, lines 113- 117: "These age- and sex-specific RRs were largely in line with the overall European RR of 1.64 recently estimated by the Global BMI Mortality Collaboration (Global BMI Mortality Collaboration et al., 2016). The differences across age groups found in that study were similar with those reported in our findings (i.e., higher RRs at younger than at older ages), though they were less distinct (Global BMI Mortality Collaboration, 2016)".

Comment 2: The effect of obesity on mortality is mediated through chronic diseases, such as cardiovascular disease. In the recent decade, there is an improvement in the treatment of cardiovascular disease, and more individuals are living with cardiovascular disease and contributing to increased life expectancy. Therefore, the contribution of obesity to mortality could be smaller due to better treatment of chronic diseases. I think authors should discuss in the discussion to explain the smaller estimates compared to the previous study (Preston & Stokes, 2011).

Reply

A decrease in cardiovascular mortality has indeed been observed in recent years, partly as a result of improvements in medical care (Mehta 2011). This development could indeed exert an influence on the association of obesity with mortality in terms of relative risks. Previous studies that assessed changes over time in the association of obesity with mortality did so only for the US, and, unfortunately, provided mixed evidence, with some of these studies reporting a decline (Flegal et al., 2005; Mehta & Chang, 2011; Yu, 2012), and others finding an increase (Yu, 2016). As there are no existing studies that have investigated these issues in Europe, whether and, if so, how the RRs of dying from obesity are changing over time in Europe is as yet unknown. Thus, we do not know how such a development would influence our results. We currently stress this point on page 14, lines 268-278: "In addition, as time-variant European RRs were not available, we had to apply time-constant RRs, even though it is possible that changes in the association of obesity with mortality – which could, for example, occur because of improvements in the treatment of chronic diseases – have affected the impact of obesity on life expectancy. Previous studies that assessed changes over time in the association of obesity with mortality did so only for the US, and, unfortunately, provided mixed evidence, with some of these studies reporting a decline (Flegal et al., 2005; Mehta & Chang, 2011; Yu, 2012), and others finding an increase (Yu, 2016). Therefore, before implementing time-variant European RRs, more information on their direction is required".

We do not believe that this issue can explain why the differences in our estimates were smaller than those in the estimates of Preston et al. (2011). While it is true that the two studies used different RR and prevalence data, if RRs had been the source of the differences between our estimates and Preston's estimates, then the observed differences would have had the same direction for all countries – which was not the case (see the manuscript's Supplementary Material, Table S4). Instead, we believe that the observed differences are primarily attributable to the different prevalence data used. We emphasised this point in our comparison of the results, page 15, lines 300-302: "Given that the observed differences do not have the same direction for the different countries, we believe that these differences are mainly attributable to the prevalence data used".

Comment 3: What is the contribution of the differences in socioeconomic status across 26 European countries, especially comparing East and West Europe to the results?

Reply

On page 15 of our manuscript, we stated that differences in socio-economic conditions, especially between Western and Eastern European populations, could help to explain why the obesity epidemic has progressed differently in these two European regions, as was previously observed in the literature (Bray & Bouchard, 2003; Finucane et al., 2011; Silventoinen et al., 2004) .

Consequently, these differences in socio-economic conditions could also be linked to the finding that the PGLS estimates are higher in Eastern Europe than in Western Europe. However, calculating the impact these differences in socio-economic conditions would have on our results is beyond the scope of our analysis.

Nonetheless, we realised that in our discussion, we focused only on explaining the East-West differences in the results for the changes over time, and not for the current PGLÉ estimates. We have now extended this explanation on page 17, lines 338- 347: "The overall progress of the obesity epidemic was lower in CEE than in Western Europe, and the increase was not constant (Finucane et al., 2011). Indeed, in CEE, increases in obesity prevalence (Bray & Bouchard, 2003; Finucane et al., 2011), OAMFs, and PGLÉ stagnated in the 1980-2008 period, and were more pronounced in the 1990s (Bray & Bouchard, 2003; Finucane et al., 2011). However, as these countries started the study period with higher obesity prevalence levels, these trends resulted in the CEE having higher average PGLÉ levels than the West. The pattern of stagnation observed in the CEE could be explained by the decrease in energy supplies at the beginning of the 1990s in CEE (Silventoinen et al., 2004) resulting from the dramatic economic and political changes in those countries (Bray & Bouchard, 2003; Finucane et al., 2011; Silventoinen et al., 2004), and which in turn affected the socio-economic status of the population (SES)".

Reviewer: 3

Reviewer Name: Paola Zaninotto

Institution and Country: UCL

Please state any competing interests or state 'None declared': None declared

Please leave your comments for the authors below

Reviewer's Comments to Author

Dear Editorial board,

I have read the paper and although I thought that I could provide my expertise in revising it, I have realised that the methods used are not familiar to me, and although I tried to read about them I felt that the statistical analysis was not sufficiently described for me to assess the appropriateness.

I feel that a demographer might be better placed in reviewing this article.

I am really sorry for not being able to help as much as I would have liked on this occasion.

Kind regards

Paola

Reviewer: 4

Reviewer Name: Haomiao Jia

Institution and Country: Columbia University, USA

Please state any competing interests or state 'None declared': None declared

Reviewer's Comments to Author

This study assessed the population level burden of disease associated with obesity by estimating potential gains in life expectancy by eliminating obesity-attributable mortality from all-cause mortality for 27 countries from 1975 to 2012.

This study used data from different sources for the calculations: (1) prevalence of obesity (BMI \geq 30kg/m²), by age, sex, country, and year; (2) relative risk (RR) of dying from obesity, by age categories and sex; and (3) all-cause mortality, by age, sex, and year.

The method for the calculation of potential gains in life expectancy is a valid method. The validity and reliability of estimates depend on the data used in this study. More detailed description of data sources is needed.

Reply

We clarified our description of the data sources.

Comment 1: All-cause mortality estimation: All-cause mortality data by age and sex were used for the estimation of life expectancy (life table method). The authors said that this is "single year of age, sex, and year". The authors did not mention whether this is by counties or combined data for all counties. If this is for each counties, what is reliability of the data? Particularly for some small counties. If this is combined data for all counties, further discussion of this weakness is needed. Since life expectancy varied greatly across European counties (approximately 10 years from the lowest to the highest), I wonder if you can use combined data.

Reply

We used the data by country, and clarified this point in the text on page 6, lines 123-124: "All-cause mortality numbers and exposure population data by single year of age, sex, year, and country were obtained from the Human Mortality Database (16).

We also clarified the high quality of these data on page 6, lines 124-126: "These data are of high quality, and are widely used within the demographic community and beyond (Barbieri et al., 2015)".

Comment 2: Obesity prevalence estimates: What data were used for the estimation? The estimates were model based. Different models for different counties, or a single model for all counties? What are predictors for these estimates?

Reply:

The obesity prevalence estimates used in our study come from the NCD Risk Factor Collaboration study (NCD Risk Factor Collaboration, 2016). As inputs for the obesity estimates, measured height and weight data from representative data sources were used. In total, 1698 population-based measurement studies with 19.2 million participants were used. These data were entered into a Bayesian hierarchical model, which also included as explanatory variables national income,

proportion of population living in urban areas, mean number of years of education, and summary measures of the availability of different food types for human consumption. This model was applied to all countries. We clarified this point in the data section, page 5-6, lines 102-109.

Although we already evaluated the use of these data in our discussion, we have now added table S3 in the Supplementary Material of the manuscript (see also Appendix Table 1 of the current reply), which gives the confidence intervals around the age-standardised prevalence estimates for each country by sex, in order to provide more information on the relative reliability of the data for the different countries in our analysis.

We also included a comment on these confidence intervals on page 13, lines 254- 261: “For those countries with less available obesity data – especially the CEE countries – a portion of the data we used were merely the result of modelling. Thus, the resulting estimates should be treated with some caution (NCD Risk Factor Collaboration, 2016). By contrast, for the non-CEE countries, most of the data we used pertained to measured data. Supplementary Material, Table S3 gives the confidence intervals around the age-standardised prevalence estimates for each country by sex in order to provide more information on the relative reliability of the data for the different countries in our analysis”.

Comment 3: RR of dying from obesity: Is obese persons relative to not obese persons or obese persons relative normal weight persons? Many studies estimated relative risk (and sometimes, hazard ratio) of dying of obese persons relative to normal weight persons. If RR is obese persons relative to normal weight person, the estimates are not accurate. Please confirm that RR is relative to not obese persons.

Reply

In line with previous studies that estimated obesity-attributable mortality (Allison et al., 1999; Banegas et al., 2003; Flegal et al., 2005; Katzmarzyk & Ardern, 2004a), we used the RRs from a meta-analysis, which included studies that used the normal weight group (BMI 18.5-24.9 kg/m²) or a narrower range of the normal weight group as a reference group (Lobstein T 2010). The estimation of obesity-attributable mortality with such a RR can be considered the theoretically maximally possible attributable mortality (GBD 2017 Risk Factor Collaborators, 2018).

Instead of simply mentioning this point in the supplementary material, we have now clarified it in the main text as well, on page 6, lines 110-113.

Such an estimation of obesity-attributable mortality, using RRs with the normal weight category as a reference group, is in line with the hypothetical situation we employed when calculating the potential gains in life expectancy, in which we hypothesized the complete elimination of obesity-attributable mortality. Our PGLE estimation is in line with previous estimations (Preston et al. 2011).

We clarified this point on page 6, lines 115- 122: “In addition, the use of RRs with the normal weight category as the reference category is in line with previous studies that estimated obesity-attributable mortality (Allison et al., 1999; Banegas et al., 2003; Flegal et al., 2005; Katzmarzyk & Ardern, 2004b),

while the estimation of obesity-attributable mortality with such a RR can be considered the theoretically maximally possible attributable mortality (GBD 2017 Risk Factor Collaborators, 2018)".

Comment 4: Another weakness is applying same RRs to all 27 countries from 1975 to 2012. Many factors (including race/ethnicity and geographic regions) were associated with the impact of obesity on mortality. It might be better applying different RRs for different countries (at least for select countries where such data are available).

Reply

This is indeed a limitation of our study, as we briefly discussed in our evaluation of data and methods section (page 13), lines 262-268. We have extended our discussion of this limitation as follows: "Because age- and sex-specific RRs of mortality associated with obesity are not readily available by country and year, we have decided to apply to all of the countries studied age- and sex-specific RRs from Western European and US populations that are largely suitable for our setting, as has previously been done (Preston & Stokes, 2011). Although RRs could differ slightly across contexts, studies that compared RRs across continents found only small differences in RRs between Europe and North America (Global BMI Mortality Collaboration et al., 2016). Consequently, we do not expect to observe large differences between individual countries".

Other

Comment 5: Reliability of estimates: This study did not provide standard error or confidence limits of estimates. This information might be important and should be reported (see my comment on reliability life expectancy estimation for small countries above).

Reply

It is indeed the case that we did not provide standard error or confidence limits for our estimates. The main reason why we did not do so is that the relative risks we used were not accompanied by uncertainty estimates. As we had uncertainty estimates for obesity prevalence only, we were unable to perform a formal analysis that would have provided uncertainty estimates that captured the full level of uncertainty. We have now clarified this point in the manuscript on page 14, lines 283-285: "The lack of information on the uncertainty of the RRs we used limited us in estimating confidence intervals for the OAMFs and PGLEs".

To give the reader some idea of the level of uncertainty, we now provide in Table S2 in the Supplementary Material of the manuscript the uncertainty estimates of the age-standardised obesity prevalence for the latest year (2012); estimated using the confidence intervals of obesity prevalence that we obtained from the data providers (NCD Risk Factor Collaboration, 2016) (See also Table 1 in the Appendix of this document). These CIs indicate that the levels of uncertainty around the estimates were somewhat greater among Central Eastern European countries than among Western European countries.

To provide some information on how the uncertainty of the prevalence estimates affected our PGLE estimates, we calculated the lower bound and the upper bound of the PGLE for 2012, for which only the levels of uncertainty of obesity prevalence estimations were taken into account (see Table 2 in the Appendix of this document). However, we decided not to include these estimates in our manuscript because they do not capture the full degree of uncertainty of the estimates, and could therefore be misinterpreted.

Comment 6: Figures are different to see.???

Reply

We checked our figures for clarity, readability, and similarity of layout, and made some improvements.

Appendix

Figure 1: Smoothed age and sex-specific RRs by single year of age (18-100)*

* by applying linear regression to the RRs by specific age groups (<50, 50-59, 60-69 and ≥70 years) which were obtained from Lobstein et al.

Table 1: Age-standardised obesity prevalence and 95% Confidence intervals, in 26 European countries (differentiating Western and Central Eastern Europe) and USA, 18-100 years in 2012.

Country	Age-standardised (stand.) obesity prevalence (%)			
	Age-stand. prevalence	95% Confidence intervals	Age-stand. prevalence	95% Confidence intervals
	Men		Women	
Central Eastern				
Belarus	20.4	13.6; 28.5	25.4	17.7; 34.1
Czech Republic	25.6	18.9; 33.4	25.2	18.3; 33.0
Estonia	21.0	16.0; 26.8	22.7	17.4; 28.8
Hungary	24.6	17.9; 32.0	22.1	15.4; 29.8
Latvia	22.2	15.2; 30.2	25.8	18.4; 34.5
Lithuania	23.8	16.9; 31.8	28.7	21.2; 37.1
Poland	23.6	18.0; 29.7	25.7	19.5; 32.5
Russian	20.4	14.8; 26.8	29.1	22.6; 36.2
Slovakia	22.2	15.7; 29.6	22.2	15.8; 29.6
Ukraine	17.7	11.2; 25.7	23.9	16.3; 32.5
Western Europe				
Austria	21.1	14.7; 28.2	19.1	13.4; 25.5
Belgium	22.9	17.3; 29.1	22.7	17.1; 28.9
Denmark	21.3	15.5; 27.9	18.4	13.1; 24.5
France	22.6	16.3; 29.6	23.1	16.9; 29.9
Finland	22.3	17.1; 28.2	21.7	16.7; 27.3
Ireland	26.2	19.4; 33.8	26.0	19.5; 33.2
Iceland	22.3	15.6; 29.8	21.0	14.7; 28.3
Italy	22.3	17.0; 28.1	23.5	18.1; 29.6
Luxembourg	24.9	17.6; 32.9	20.7	14.3; 28.0
Netherlands	18.9	13.8; 24.5	20.3	15.3; 25.8
Norway	24.5	18.5; 31.2	23.7	18.0; 30.2
Portugal	19.6	13.8; 26.4	21.1	15.0; 28.1
Spain	24.2	18.5; 30.5	26.4	20.3; 32.8
Sweden	21.6	16.3; 27.4	19.8	14.7; 25.6
Switzerland	22.0	16.3; 28.3	18.6	13.1; 24.8
United Kingdom	26.5	22.3; 31.2	29.1	24.8; 33.6
USA	33.4	27.5; 39.5	35.5	29.7; 41.5

Table 2: Potential gains in life expectancy at birth (PGLE) (upper and lower bound included) if obesity-attributable mortality was eliminated, in 26 European countries (differentiating Western and Central Eastern Europe) and the USA, 2012

Country	PGLE 2012					
	PGLE	PGLE	PGLE	PGLE	PGLE	PGLE
		lower	upper		lower	upper
	Men			Women		
Central Eastern Europe (CEE)						
Belarus	1.41	0.95	1.92	1.19	0.82	1.63
Czech Republic	1.39	1.00	1.83	1.03	0.71	1.41
Estonia	1.37	1.04	1.76	1.04	0.75	1.37
Hungary	1.52	1.11	1.96	1.04	0.71	1.44
Latvia	1.48	1.02	1.99	1.18	0.81	1.60
Lithuania	1.67	1.20	2.22	1.31	0.94	1.73
Poland	1.48	1.12	1.87	1.19	0.88	1.54
Russian Federation	1.53	1.13	1.98	1.54	1.19	1.90
Slovakia	1.31	0.92	1.76	0.06	0.64	1.32
Ukraine	1.25	0.80	1.78	1.16	0.78	1.59
Western Europe						
Austria	1.03	0.68	1.42	0.73	0.48	1.05
Belgium	1.17	0.86	1.52	0.97	0.69	1.29
Denmark	1.04	0.71	1.43	0.79	0.52	1.12
France	1.18	0.82	1.60	0.84	0.57	1.18
Finland	1.19	0.88	1.54	0.90	0.64	1.21
Ireland	0.97	0.62	1.41	0.80	0.50	1.17
Iceland	1.21	0.85	1.63	1.01	0.70	1.36
Italy	1.06	0.78	1.38	0.93	0.68	1.22
Luxembourg	1.19	0.79	1.65	0.79	0.50	1.14
Netherlands	0.86	0.60	1.15	0.88	0.64	1.16
Norway	1.07	0.76	1.44	0.91	0.64	1.22
Portugal	1.01	0.67	1.40	0.81	0.53	1.15
Spain	1.22	0.89	1.58	1.05	0.74	1.38
Sweden	0.91	0.64	1.23	0.76	0.52	1.07
Switzerland	0.93	0.63	1.30	0.66	0.42	0.96
United Kingdom	1.27	1.03	1.53	1.09	0.89	1.32
USA	1.73	1.39	2.10	1.44	1.15	1.76

References

- Dagfinn A, Abhijit S, Manya P, Teresa N, Imre J, Serena T et al. BMI and all cause mortality: systematic review and non-linear dose-response meta-analysis of 230 cohort studies with 3.74 million deaths among 30.3 million participants *BMJ* 2016; 353 :i2156
- Alley DE, Lloyd J, Shardell M. Can obesity account for cross-national differences in life expectancy trends? In: Crimmins EM, Preston SH, Cohen B, editors. *International differences in mortality at older ages: Dimensions and sources, panel on understanding divergent trends in longevity in high-income countries*. Washington, DC: National Academies Press; 2011. p. 164-92.
- Barbieri M, Wilmoth JR, Shkolnikov VM, Gleit D, Jasilionis D, Jdanov D, et al. Data Resource Profile: The Human Mortality Database (HMD). *Int J Epidemiol*. 2015 Oct;44(5):1549-56.
- Flegal KM, Kit BK, Orpana H, & Graubard BI (2013). Association of all-cause mortality with overweight and obesity using standard body mass index categories: A systematic review and meta-analysis. *JAMA*, 309(1), 71-82.
- Flegal KM, Graubard BI, Williamson DF, Gail MH (2005). Excess deaths associated with underweight, overweight, and obesity. *JAMA*. 2005 Apr 20;293(15):1861-7.
- GBD 2017 Risk Factor Collaborators (2018). Global, regional, and national comparative risk assessment of 84 behavioural, environmental and occupational, and metabolic risks or clusters of risks for 195 countries and territories, 1990-2017: a systematic analysis for the Global Burden of Disease Study 2017. *Lancet*, 10;392(10159):1923-94.
- Global BMI Mortality Collaboration, Di Angelantonio E, Bhupathiraju S, Wormser D, Gao P, Kaptoge S, et al (2016). Body-mass index and all-cause mortality: individual-participant-data meta-analysis of 239 prospective studies in four continents. *Lancet*, 20;388(10046):776-86.
- Mehta, N. K., & Chang, V. W. (2011). Secular declines in the association between obesity and mortality in the United States. *Population and Development Review*, 37(3), 435-451.
- Yu, Y. (2012). Reexamining the declining effect of age on mortality differentials associated with excess body mass: Evidence of cohort distortions in the United States. *American Journal of Public Health*, 102(5), 915-922.
- Yu, Y. (2016). The changing body mass-mortality association in the United States: Evidence of sex-specific cohort trends from three national health and nutrition examination surveys. *Biodemography and Social Biology*, 62(2), 143-163.
- Lobstein T, Leach RJ.(2010). *Workpackage 7: Overweight and Obesity Report on data collection for overweight and obesity prevalence and related relative risks*. 2010.
- Preston SH, Stokes A.(2011). Contribution of obesity to international differences in life expectancy. *American Journal of Public Health*,101(11):2137-43.

VERSION 2 – REVIEW

REVIEWER	Lena BJörck University of Gothenburg, Sweden
REVIEW RETURNED	06-May-2019

GENERAL COMMENTS	My comments have been adressed and the paper is now suitable for publication.
---

REVIEWER	Klodian Dhana Rush University Medical Center, United States
REVIEW RETURNED	20-May-2019

GENERAL COMMENTS	Thank you for addressing my comments. I have no addition question.
--

REVIEWER	Haomiao Jia Columbia University U.S.A.
REVIEW RETURNED	06-May-2019

GENERAL COMMENTS	The authors have satisfactorily responded to all my questions and made the necessary changes to the manuscript.
---